# Decolorization of azo dyes by a novel aerobic bacterial strain Bacillus cereus strain ROC

Anum Fareed[1], Habiba Zaffar[2], Muhammad Bilal[2], Jamshaid Hussain[1], Colin Jackson[3], Tatheer Alam Naqvi [1]*

1 Department of Biotechnology, COMSATS University Islamabad, Abbottabad, Pakistan, 2 Department of Environmental Sciences, COMSATS University Islamabad, Abbottabad, Pakistan, 3 Research School of Chemistry, Australian National University, Canberra, Australia

* tatheer@cuiatd.edu.pk, statheer_2000@hotmail.com

**Data Availability Statement:** All relevant data are within the paper and its Supporting Information files.

**Funding:** The authors received no specific funding for this work.

## Abstract

Synthetic dyes are widely used as colorant compounds in various industries for different purposes. Among all the dyestuffs, azo dyes constitute the largest and the most used class of dyes. These dyes and their intermediate products are common contaminants of ground water and soil in developing countries. Biological methods have been found to be promising for the treatment and degradation of these compounds. In the present study, we focused on the biological removal of azo dyes (Reactive orange 16 and Reactive black 5) under aerobic conditions using an indigenous bacterial strain isolated from contaminated industrial areas. The bacterial isolate was identified as *Bacillus cereus* strain ROC. Degradation experiments under agitation with both free and immobilized cells indicates that this strain degrades both azo- dyes in 5 days. The immobilized cells were more proficient than their free cell counterparts. The toxicity of the biotransformation products formed after decolorization were assessed by conducting bacteriotoxic and phytotoxic assays. All the toxicity assays indicate that the dyes' degraded products were non-toxic in nature, as compared to the dyes themselves. The kinetics of the azo dyes' degradation was also studied at various initial concentration ranges from 50 mg/L to 250 mg/L by growth independent kinetic models. Zero-order kinetics were fit to the experimental data, producing values of least squares regression ($R^2$) greater than 0.98, which indicates that the bacterial strain degrades both dyes by co-metabolism rather than utilizing them as sole energy source. These results indicate that the *Bacillus cereus* ROC strain has great potential to degrade dye-contaminated water and soil.

## Introduction

Dyes are colored compounds (often synthetic) and are extensively used in pharmaceutical, leather, textile, cosmetics, food and paint manufacturing industries [1]. The annual global production of synthetic dyes is approximately 700,000 tons [2]. Dyes can be broadly classified according to their chemical structures, applications, and origin i.e., natural, or synthetic [3]. Based on their chemical structures, they can be broadly classified as triphenylmethane, anthraquinone, phthalocyanines and azo dyes. Furthermore, they can be classified according to their usage as anionic, non-ionic, and cationic [4]. A dye is generally composed of one or more

**Competing interests:** The authors have declared no competing interest.

**Abbreviations:** RO-16, Reactive Orange 16; RB-5, Reactive black 5; IMC, Immobilized cells; mg/L, milligram per litre; HPLC, High performance liquid chromatography; ppm, part per million; mM, milli mole; μl, micro litre.

benzene rings called chromogens, chromophores, and auxochromes [5]. The chromophore, also known as the color bearing group, is responsible for giving color to a dye, whereas auxochromes strengthen the color of chromophores and make them soluble in water [6].

Azo dyes are the most used class of dyes, constituting more than half of all the dyes produced [7]. The chemical structure of these dyes is characterized by the presence of azo bonds (-N = N-), connected with naphthalene or benzene rings [1]. These aromatic rings can have different substituents such as sulfonic acid ($SO_3H$), chloro (-Cl), hydroxyl (-OH), methyl (-$CH_3$), carboxyl (-COOH) nitro (-$NO_2$) and amino ($NH_2$) groups [6]. These substituents make azo dyes the most structurally flexible and diverse group of dyes by making them water soluble and resistant to degradation under environmental conditions [8]. The use of azo dyes is preferred in textile industries because of their economically attractive features such as easy application, strong covalent binding to textile fibers, high photolytic structural stability, the availability of a variety of shades, and minimal energy requirements for their production [7].

Due to increased industrialization and the increasing use of colors in the textile and clothing industry, dyes are frequently discharged into the environment in large quantity and therefore constitute an important component of wastewater effluent, especially in developing countries where most of the textiles are produced [9]. During the manufacturing process, approximately 10–15% of the total amount of these dyes is released into the environment via textile wastewater [1]. These dyes are difficult to decolorize because of their complex structures and they bio-accumulate in ecosystems, causing serious health problems [9]. The presence of these dyes can cause adverse effect on all the organisms present in an ecosystem, as these dyes and their intermediate products (e.g. benzidine [1,1'-biphenyl-4,4'-diamine], 1-amino-2-naphthol, 6-amino naphthalene sulfonic acid, o-tolidine) are highly mutagenic and carcinogenic in nature [10]. In humans and animals, they cause bladder cancer, skin dermatoses, allergic responses, eczema, and have adverse effects on the lungs, liver, circulatory, reproductive, and immune system [11]. These dyes also affect the rate of photosynthesis in aquatic life by decreasing the intensity of light penetration and have been found to be toxic to aquatic organisms including animals and plants due to the formation of intermediate aromatic compounds [10].

Several physical and chemical methods such as precipitation, adsorption, filtration, coagulation, oxidation, electrochemical destruction, ozonation, and photo-catalysis have been used for the destruction of dyes, but these methods generally involve complicated procedures and are not economically feasible [12]. Microbial decolorization and degradation of these dyes is cost effective, efficient and an ecofriendly alternative to physical and chemical decomposition [13]. The usefulness of microbial processes for the biodegradation of these dyes depends on the activity and the adaptability of the selected microbes. The activity and adaptability of the microorganisms towards the recalcitrant pollutants depends upon the activities of the enzymes that they synthesize [12,13].

Many microorganisms such as algae, yeast, fungi, and bacteria, have been reported to decolorize and degrade azo dyes. However, most of these studies describe the decolorization of azo dyes under anoxic conditions, leading to the formation of colorless toxic metabolites. On the other hand, aerobic degradation offers a comparatively easier method for azo dye degradation. While aerobic degradation holds great promise, there is still a need to assess the extent of degradation of the toxic intermediate products formed by these dyes under aerobic conditions. In this study, we describe the isolation and identification of a bacterial strain with the capacity to degrade azo dyes i.e., Reactive black 5 and Reactive orange 16 under aerobic conditions. The effects of different environmental conditions i.e., temperature, pH and the concentration of dye were also studied to fully characterize the biological degradation of these molecules. Furthermore, cell immobilization was used to study the rate of dye decolorization in comparison

with freely suspended bacterial cells. Cell immobilization was found to protect the cells from the lethal effects of dyes and their by-products, maintain high biomass concentration, was cost-effective and re-usable [14]. The biodegradation kinetics of the azo dyes with respect to the degradation potential of microbial communities has not been studied in detail, yet can provide a deeper understanding of the specific biodegradation mechanisms. Thus, the present study investigated the decolorization efficiency of the bacterial strain at various initial dyes concentrations using zero order, first order and second order growth independent models [15]. Finally, phytotoxicity and biotoxicity assays were also performed to assess the toxic nature of the dissolved metabolites formed by the biotransformation of RB-5 and RO-16.

## Research methodology

### Chemicals and microbiological media

All chemicals and reagents of analytical grade with 99% purity unless otherwise mentioned. Azo dyes i.e., RB-5 and R0-16 were purchased from Sigma Aldrich, USA. The stock solutions (1000 mg/L) of each dye were prepared by dissolving 1 g of dye in 1 L HPLC grade water. The working solutions of each dye were prepared by diluting the stock solutions and the initial pH were adjusted by using HCl and NaOH solutions (1N). Nutrient broth and nutrient agar were used as enrichment and isolation media.

### Isolation, screening, and identification of azo dyes' degrading bacteria

Soil samples were collected from the disposal areas of textile-based industries in Faisalabad and its adjacent locations mainly the Paharan drain carrying textile effluents which is situated at a latitude of 31.476677˚ and longitude of 73.070585˚. Moreover, the site is not the private entity of any person or group or industry and does not contain any endangered or protected species therefore, there is no need of any special permit for sampling. All the samples were collected aseptically and were preserved at 4˚C until use. To obtain the dye-degrading bacterial strains, 10 g of soil sample was inoculated into 100 mL of LB medium supplemented with 25 mg/L of each respective dye i.e., Reactive black 5 and Reactive orange 16. The flasks were incubated under aerobic conditions with aeration and agitation at 37˚C. After 3 days of incubation, an equal amount of each dye i.e., 25 mg/L was added to the flasks followed by incubation under similar aerobic conditions. The successive acclimatization of the enriched microbial consortium was carried out till the final concentration of dyes reached to 250 mg/L [12]. The adapted microbial culture was serially diluted and 50 μl suspension was then spread on nutrient agar plates containing azo dye (100 mg/l).

Screening was performed to find out the most efficient bacterial strain capable of decolorizing both dyes using LB medium containing different concentrations of each dye. The bacterial strain that showed prominent growth on the highest concentration was selected to carry out further research.

### Strain identification and phylogenetic analysis

Bacterial strain was identified by 16S rRNA technique. The 16S rRNA genes were amplified with a pair of universal primers i.e 1492R (5′ -TACGGYTACCTTGTTACGACTT- 3′) and 27F (5′ -AGAGTTTGATCMTGGCTCAG- 3′) using 2720 Thermal cycler of Applied Biosystems by following the protocol as mentioned by Pushpa et al., (2019) [16]. The amplified sequence was subjected to nucleotide blast to find similarity between existing sequences available in NCBI data base. MEGA 7 software was used to construct the phylogenetic tree based on obtained blast results.

## Agar entrapment of bacterial cells

Bacterial cells were immobilized by using the agar entrapment method [15]. Briefly, 0.1 g/ml agar was dissolved in 0.9% NaCl solution and sterilized for 15 min at 121˚C. After cooling the molten agar solution (15 ml) was mixed with cell slurry and poured into sterilized petri plates. The solidified agar containing bacterial cells was cut into small cubes and placed in 100 mM phosphate buffer (pH 7.5) and was cured at 4˚C for 1 h. The buffer was removed after curing immobilized cells were preserved at 4˚C till further use.

## Dye decolorization experiments and quantitative analysis

Dye decolorization experiments were carried out using the free and immobilized cells of the bacterial strain ROC at different concentrations of Reactive black 5 and Reactive orange 16, i.e., 0, 50, 100, 150, 200 and 250 mg/L. Before performing decolorization experiments, the environmental factor i.e., pH was optimized for bacterial growth. The bacterial strain was incubated in LB medium with wide range of pH (5–11). Afterwards, the dye decolorization experiments were conducted at pre-optimized pH (7) while incubated at various temperatures i.e., 25, 32, 37 and 45 ˚C with different concentrations of each dye on a shake flask incubator at 220 rpm for 6 days. Each experiment was conducted individually with each temperature using both free and IMC. All experiments were conducted in replicates of three. Samples were drawn after every 24 hours for measuring the decolorization activity of the free and IMC of the strain. The samples were centrifuged at 8,000 rpm for 20 min and the residual dye left in the supernatant was measured at $\lambda_{max}$ of 597 nm using T80 UV/Vis spectrophotometer. For further quantification, the residual dyes left in control and samples were extracted with ethyl acetate [17]. The extracts were dried by adding $Na_2SO_4$ (anhydrous) and then evaporated. The dried residues were dissolved in HPLC grade ethyl acetate and were analyzed.

The decolorization activity was expressed by following Eq.

$$Percentage\ Decolorization = \left( Initial\ absorbanceFinal \frac{absorbance}{Inital} absorbance \right) * 100$$

## Phytotoxic effects

Phytotoxic effects of the parent dyes and their metabolites i.e., pre and post bioremediation samples were studied for the germination of commercially important *Solanum lycopersicum (Riogrande)* seeds. The concentrations of the dyes used were 50 ppm, 150 ppm and 250 ppm. The metabolites of both dyes (Reactive black 5 and Reactive orange 16) were extracted in ethyl acetate followed by air drying. Afterwards, the extracted metabolites were dissolved in 15 mL of distilled water. Prior to experiment, the seeds were soaked in mercuric chloride 0.1% for 5 min and then washed with distilled water. The seeds were left on filter paper (Whatman No.1) for drying. Consequently, the sterile petri plates having filter papers were moistened with pre and post treated samples and controls (distilled water). The 15 seeds were placed onto a soaked filter paper in each single plate followed by incubation at 22–24˚C under light. The seed germination was counted at 3[rd] day of incubation, while the mean value of root and shoot length was recorded after 5 days of incubation [18].

## Bacteriotoxic effects

Pre- and post-bioremediation toxicity of dyes samples was assessed by performing agar well diffusion assay as mentioned by Jamil et al., (2012) with minor modifications [19]. The assay was performed against three reference bacterial strains i.e., *Pseudomonas aeruginosa*, *Escherichia coli* and *Staphylococcus aureus*. All the reference bacterial strains were primarily enriched

via repetitive culturing in Luria Bertani (LB) medium and then incubated for three successive days in shaking incubator (220 rpm). Afterwards, an aliquot (750 µL) from each culture broth (optical density equal to 0.5 MacFarland's turbidity standard) was added to nutrient agar medium as inoculum and poured into sterile petri plates. The solidified media was then used for the preparation of wells (8mm) using sterilized cork borer. An aliquot of 100 µL from each treated and untreated dye samples were added to the wells whereas ampicillin (200 µg) and autoclaved water were also added to each plate as a negative and positive control respectively. All the plates were incubated at 37˚C for 24 hrs. On successive day, the petri plates were noticed for the appearance of zones around each well. The zone diameters were measured and recorded.

## Statistical analysis

SPSS 20 software was used for statistical analysis. The comparisons among different treatments were made using one way ANOVA and Duncan test with 95% confidence level ($p < 0.05$)

## Modeling and degradation kinetics

In the present study, the Kinetic Models (zero, first and second order) were studied using dye decolorization data by following the studies carried out by [20,31].

The following equation was used to determine Zero order kinetic rate

$$C/Co = kt$$

Whereas,
First kinetic order was studied by using following form of equation

$$LnCo/Ct = kt$$

Where (Co) is the initial concentrations of reactive black 5 and reactive orange 16 and (Ct) is concentration of residues of both the azo dyes at that time. 'k' is the degradation rate constant, and its value was determined by the slope of straight line.

The second kinetic order can be determined by applying the following equation.

$$kt = \frac{1}{C_t} - \frac{1}{C_1}$$

The value of second order kinetic constant was defined by plotting dc/dt against time [21].

The Selection of the kinetic model that offered the best fit to the experimental data was established on the value of the coefficient of determination and the significance of the linear coefficient and slope of the curve (equation parameters).

# Results and discussion

## Isolation, screening, and identification of bacterial isolate

The enrichment of the soil samples collected from the disposal areas of textile-based industries led to the isolation of an efficient dye-degrading bacterial strain, which we have called ROC in this study. The bacterial strain ROC was capable of growth on the highest concentration of both dyes (Reactive black 5 and Reactive orange 16), which was 250 mg/L. The bacterial strain was identified as *B. cereus* strain ROC by 16S rRNA technique, which exhibited 99% similarity with other *Bacillus cereus* strain 16S rRNA sequences when compared using the nucleotide BLAST (basic local alignment search tool) in NCBI. The sequence was submitted to the genbank database with accession number ON063239. A multiple sequence alignment was then

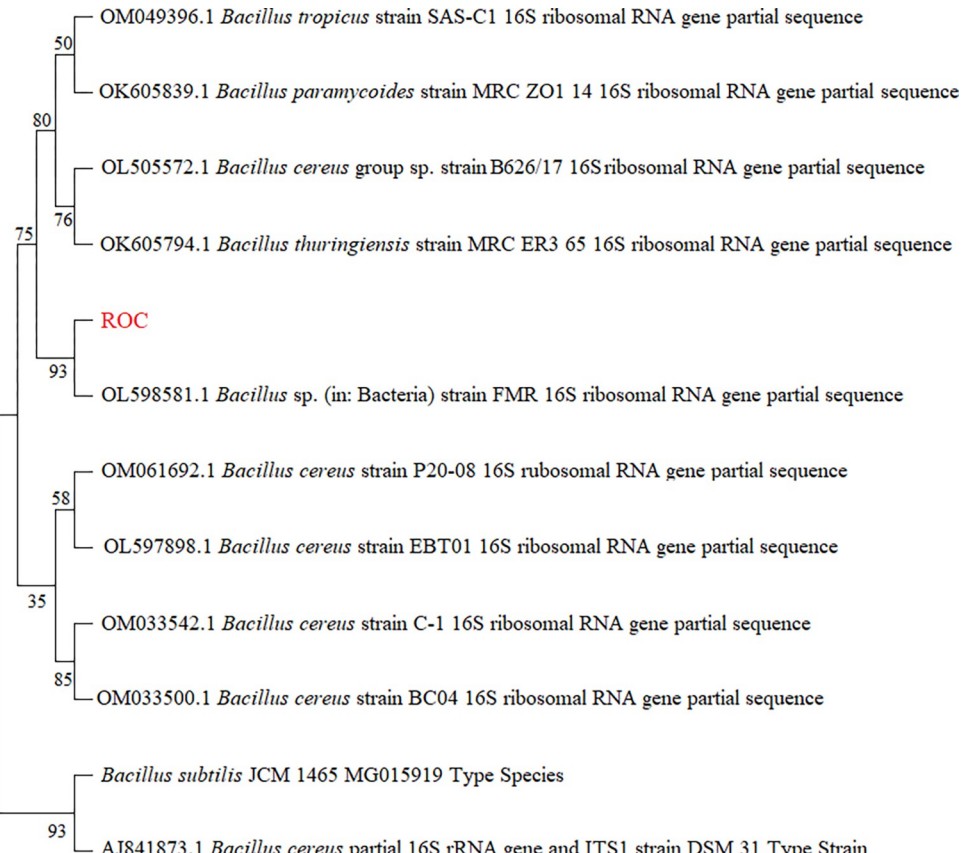

**Fig 1. Neighbor-joining phylogenetic tree of the representative bacterial strain ROC and its related species based on the 16S rRNA gene sequences.**

constructed based on blast results as shown in Fig 1, showing close homology to several other related *Bacillus* strains.

Previous studies described the degradation of azo dyes by various groups of anaerobic bacteria i.e. *Nocardiopsis alba*, *Nocardiopsis* sp. strain *dassonvillei*, *Enterococcus faecalis* sp. strain YZ 66, *Aeromonas hydrophila Pseudomonas putida*, *B. subtilis* and *Pseudomonas aeruginosa*, but aerobes are seldom reported to possess this activity [22–26]. A previous study described the biodegradation of the Reactive Orange 16 in a mixed culture of *Pleurotus ostreatus* (bacteria) and a suspension of *Candida zeylanoides* (yeast) [27]. Recently, it has been reported that a consortium of three haloalkaliphilic isolates belonged to the genus *Halomonas* exhibited a remarkable ability of decolorizing the Reactive Black 5 (87%) [28]. Previously, *Bacillus cohnii*, *Bacillus sp. YZU1* and *Bacillus pumilus* were also reported to decolorize direct dyes in anaerobic conditions [29–32].

## Immobilization and biodegradation experiments

Biodegradation experiments were conducted at various temperatures (32, 37 and 45 ˚C) using free and immobilized cells of *B. cereus* ROC and the decolorization was monitored after every 24 hours of incubation. The results indicate that the decolorization was significantly higher in immobilized cells as compared to their free-cell counter parts. In the case of immobilized cells (IMC) 100% decolorization was recorded at all the concentrations of both dyes, while ~80% decolorization was observed when free cells were employed for the same concentrations of both

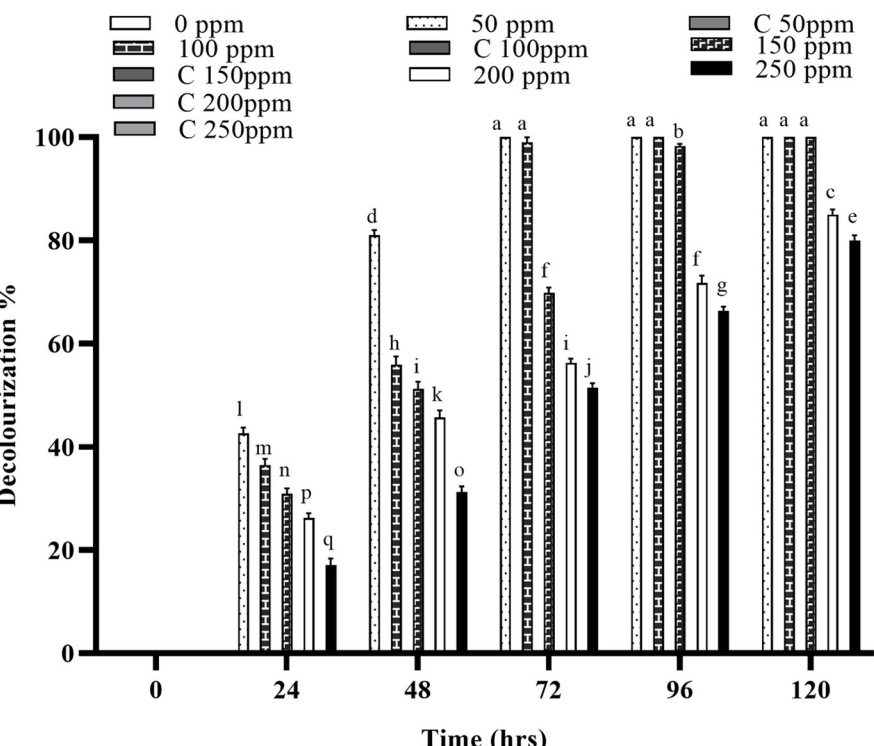

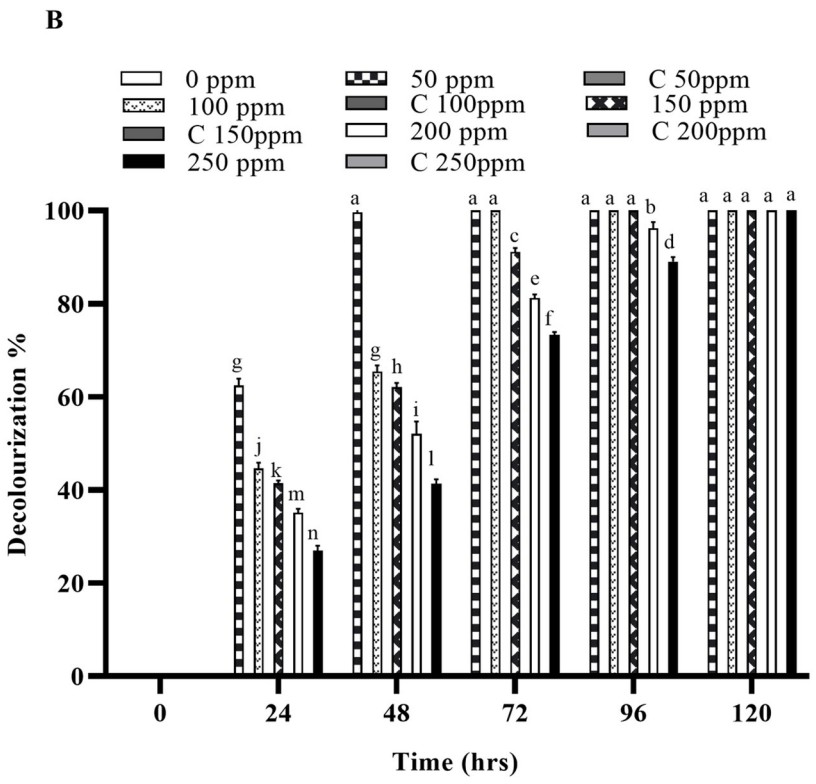

**Fig 2.** Decolorization of Reactive orange 16 by free and immobilized cells of bacterial strain (A) by free cells at 37 ˚C (B) by immobilized cells at 37 ˚C. Different alphabets show the significant differences (p < 0.05) between different treatments.

dyes. Increasing the temperature significantly increases the rate of decolorization and it was observed that the decolorization was significantly faster at 37˚C, as compared to 30˚C in case of both the free cells as well as IMC as shown in Figs 2A, 2B, 3A and 3B, S1A, S1B, S2A and S2B.

Any residual dye left in the liquid media after degradation was extracted after every 24 hours and quantified by using UV/Vis spectrophotometry. Significant degradation of these dyes was observed at 37˚C, as compared with the dyes' removal at 30˚C both by free and IMC as shown in Figs 4A, 4B, 5A and 5B, S4A and S4B. However, when the temperature was increased to 45˚C, a considerable decrease in the decolorization was observed for both free cells as well as the IMC, suggesting loss of viable cells and/or denaturation of the enzymes responsible [29,33,34]. An increased concentration of both dyes has no effect on the decolorizing ability of the bacterium but it takes more time to decolorize them, as expected. Similar findings were also observed when the temperature was increased to 45˚C with the increased dye concentrations i.e., a significant reduction in the decolorization ability of the bacterial strains. Indeed, in previous studies, *Micrococcus sp*. showed greater decolorizing ability at 37 ˚C than in higher temperatures [10]. Another report also supports our findings where the bacterial strains *Bacillus cohnii* and *Deinococcus radiodurans* lost their ability to decolorize dyes when temperature was increased beyond 37 ˚C [30].

In the present study, the IMC displayed significantly higher decolorizing ability, which might be due to their stability and greater mechanical strength (due to protection from immobilization), as compared to their free counterparts. The free cells also required sufficient growth time to acclimate in the new environment, which might be a reason of their slower rate of decolorization. Previous findings have shown that the immobilized cells were a viable option for the degradation of various hazardous compounds [25,35–38]. Similar findings were also observed in previous studies carried out on the anaerobic degradation of different dyes (Reactive Red 22, Reactive Blue 172, bromothymol blue, methylene, crystal violet, Congo red and Direct Red), where immobilized cells of the bacterial strains degraded these dyes faster than their free counterparts [32,39–41]. Another study carried out by Pandey *et al*., (2020) in which the biodegradation potential of immobilized cells of *Bacillus* sp. and *Lysinibacillus* sp. against two azo dyes i.e., Reactive Orange 16 and Reactive Blue 250 was reported [14]. Free cells of *Bacillus*. sp. decolorized 92% of RO-16, whereas its immobilized cells decolorized 97% of RO-16. Similarly, the free cells of *Lysinibacillus* sp decolorized 95% of RB 250 but on immobilization, dye decolorization occurred more rapidly i.e. 99%. The residual dyes left in the liquid media after degradation were also extracted after every 24 hours and quantified by using UV/Vis spectrophotometry.

## Estimation of kinetic parameters for dye degradation

According to the correlation coefficient values ($R^2$) obtained from dye decolorization experiments, the kinetic model that best fit to the decolorization data is a zero-order kinetic model. It is clear from the Table 1 that the $R^2$ values obtained in zero order kinetic model are higher than the values obtained in first and second-order kinetic models. The data were fit to the equation ln$Ct$/$Co$, where $Ct$ is the dye concentration at a specific time and $Co$ is the initial dye concentration, and further verified by the regression coefficient $R^2$ (Table 1). The regression coefficient ($R^2$) was calculated for each initial concentration (50, 100, 150, 200 and 250 mg/l) with both free and IMC. These results indicates that the bacterial strain did not utilize both azo dyes as their sole energy source, but rather can degrade the azo dyes by co-metabolism.

**A**

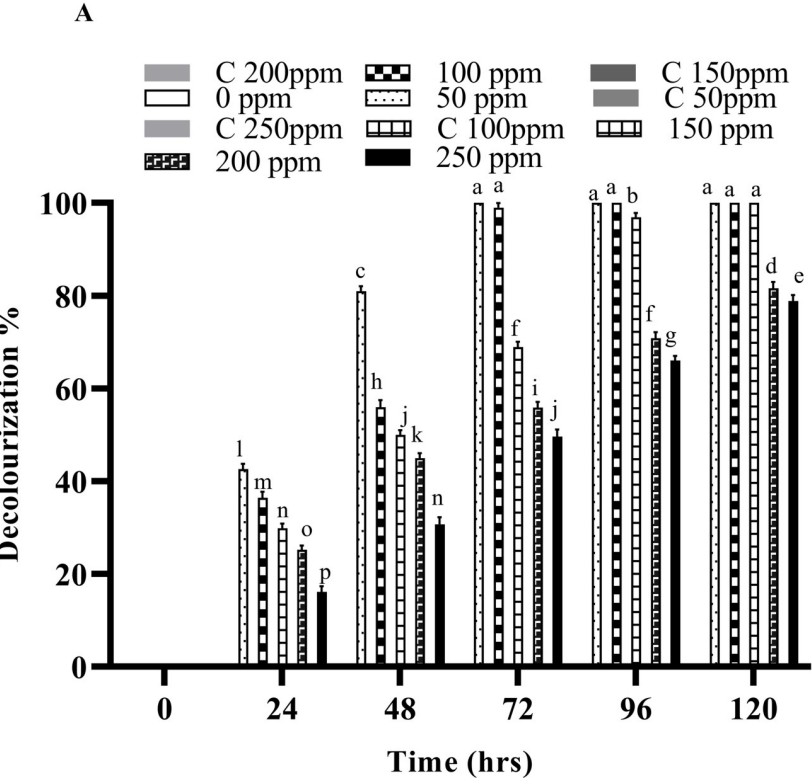

**B**

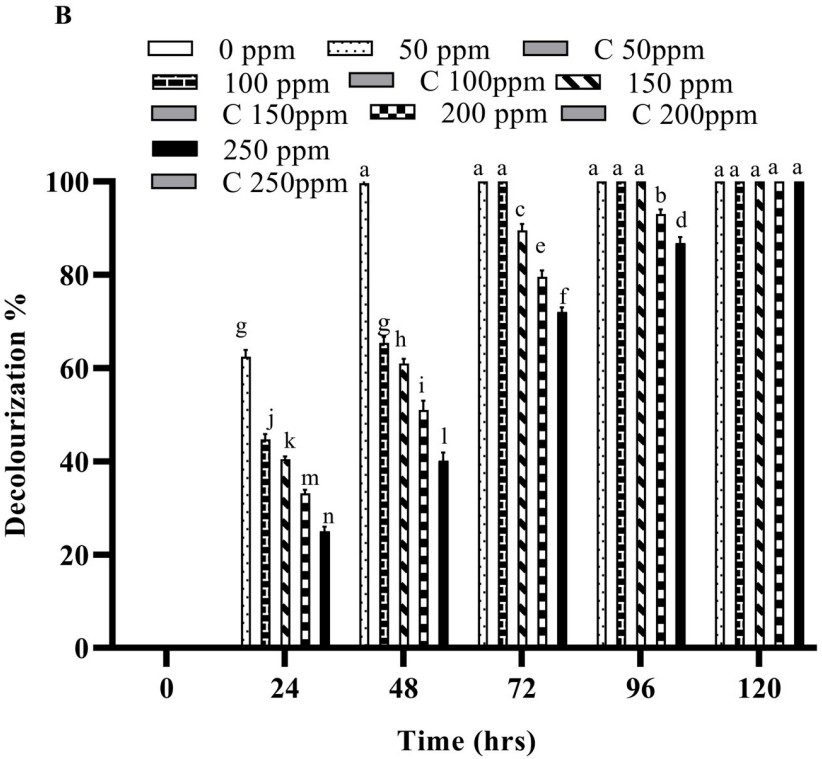

**Fig 3.** Decolorization of Reactive black 5 by free and immobilized cells of bacterial strain (A) by free cells at 37 ˚C (B) by immobilized cells at 37 ˚C. Different alphabets show the significant differences (p < 0.05) between different treatments.

## Phytotoxic effects

Various toxic natural and synthetic pollutants, including dyes, are known to inhibit the activity of plant hormones (gibberellins, auxins, and cytokinins). These phytohormones play a crucial role in seed germination as well as seedling growth [42,43]. The phytotoxic effects of both treated and untreated dyes samples generated in this study were tested for seed germination and determination of growth parameters (shoot and root length) of the commercially important plant *Solanum lycopersicum (Riogrande)*, i.e., tomato. The experimental findings were notable: the treated water, in which the dyes were decolorized by *B. cereus* ROC, exhibited a significantly increased percentage of seed germination as compared to the untreated water that retained various concentration of dyes (Reactive black or Reactive Orange; Figs 6 and 7). In this experiment, simple distilled water was used as a positive control alongside the treated water as well as untreated water containing dyes. As far as the germination of seeds is concerned, there was no significant difference observed in case of distilled water (positive control) and bacterium treated dye samples initially containing 50 ppm dye (either RB-5 or RO-16), with a percentage germination of approximately 93%. In contrast, a significant decrease in germination was observed when untreated water containing 50 ppm dye was applied to plants, where the seed germination was 85% as shown in Tables 2 and 3. Similarly, seed germination was up to 86% when treated water was used with samples of initial concentrations of 150 and 250 ppm, while in untreated dyes (with initial concentrations of 150 and 250 ppm) only 6% of seeds germinated.

The root and shoot growth were also measured in treated and untreated dye samples along with water as a positive control (Figs 6 and 7). The results revealed that the shoot and root growth was inhibited by the application of the untreated dye samples, especially at concentrations of 150 and 250 ppm, while in bacteria-treated samples the shoot and root lengths were comparable to those of plants treated with distilled water (positive control), as shown in Tables 2 and 3. This confirmed that the treated samples have no toxic effect on the growth of selected plants. It was obvious that the metabolites produced by the bacterium have no toxic effect on plant growth and hence the water treated with the bacterium can be used for irrigation purposes. A previous study conducted by Kurade et al., (2016) also reported similar findings where the metabolites obtained after decolorization of Disperse red (DR54) dye showed no toxicity as compared to the parent dye indicating the detoxification of the DR54 by *Bacillus laterosporus* [44]. Kurade et al., (2013) also found that the intermediate metabolites formed after the biodegradation of Disperse brown 118 exhibited no toxic effects on the seed germination of *R. raphanistrum* [45,46]. Prasad et al., (2013) reported the less toxic effects on the shoot and root lengths of seeds of *Vigna mungo*, *Sorghum bicolor* and *Vigna radiata* exposed to the degraded metabolites of Direct Blue-1 azo dye compared to the parent dye itself [30]. The phytotoxicity test with untreated dye (5,000 mg l$^{-1}$) showed 70% and 80% inhibition of seed germination in *V. radiata*, *V. mungo* and *S. bicolor* plants, whereas no inhibition of seed germination was observed when bacterial treated dyes carrying various metabolites were applied on these plants [34].

## Bacteriotoxic effects

The bacteriotoxic assays of treated and untreated dyes were performed on three reference bacterial strains i.e., *Pseudomonas aeruginosa*, *Escherichia coli* and *Staphylococcus aureus*. It was

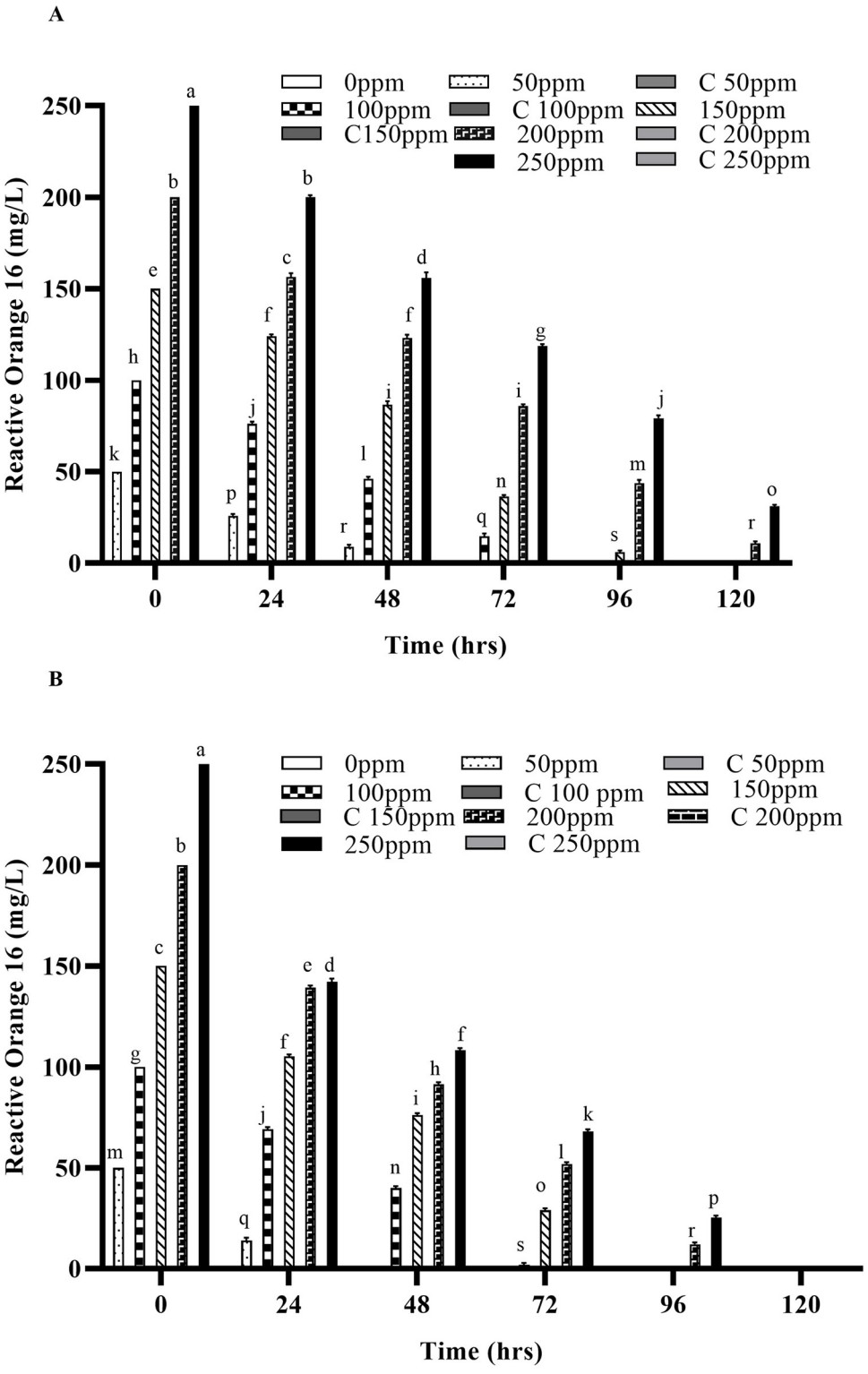

**Fig 4.** Concentrations of Reactive orange 16 degraded after treatment with free and immobilized cells of the bacterial strain (A) Degraded by free cells at 37 ˚C (B) Degraded by immobilized cells at 37 ˚C. Different alphabets show the significant differences ($p < 0.05$) between different treatments.

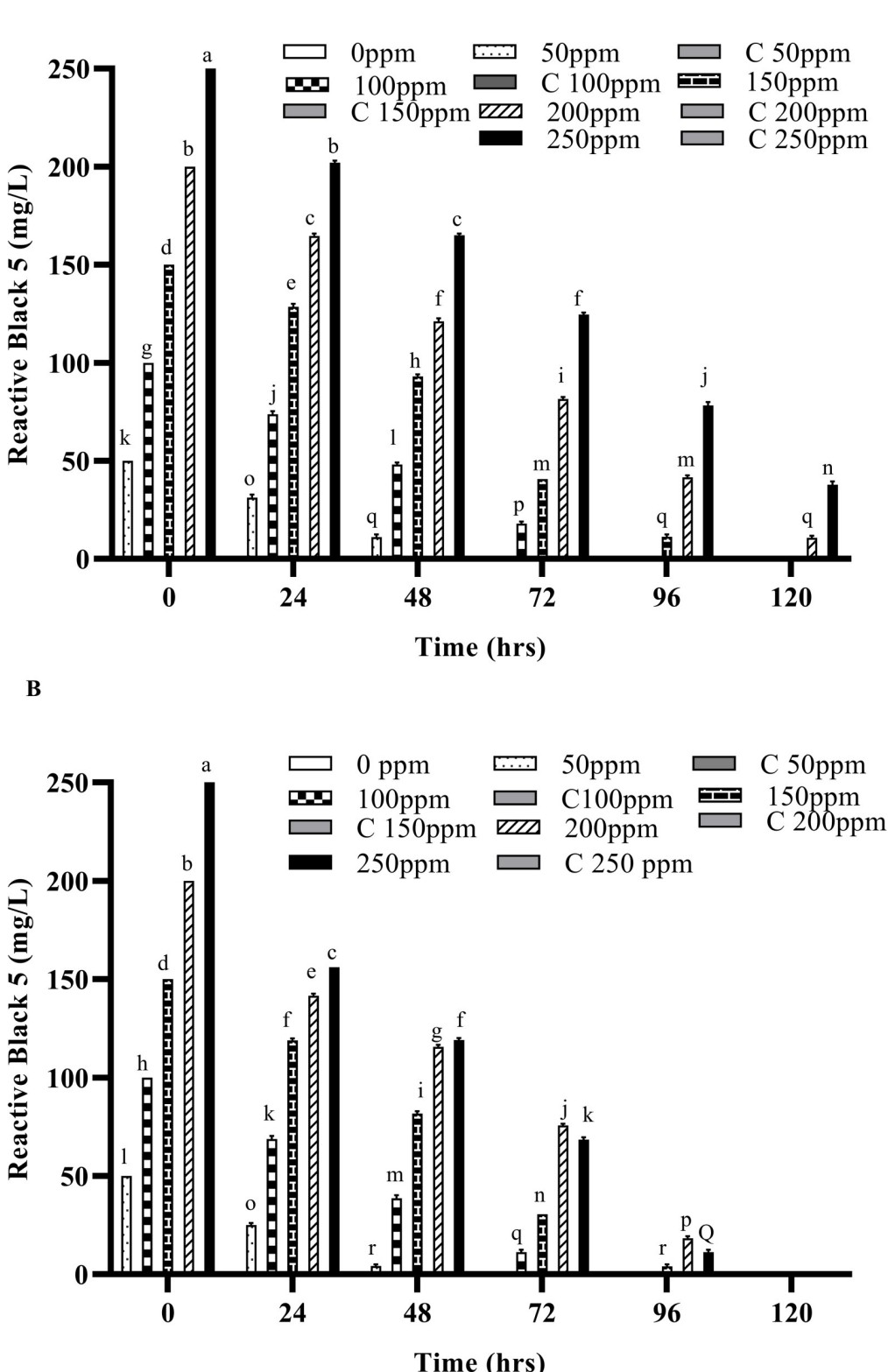

**Fig 5.** Concentrations of Reactive black 5 degraded after treatment with free and immobilized cells of the bacterial strain (A) Degraded by free cells at 37 ˚C (B) Degraded by immobilized cells at 37 ˚C Different alphabets show the significant differences ($p < 0.05$) between different treatments.

**Table 1. Growth independent kinetic models for the azo dyes' degradation at different initial concentrations.**

| Initial concentrations(mg/l) | Bacterial strain (ROC) | Zero order ($R^2$) | | First order ($R^2$) | | Second order ($R^2$) | |
|---|---|---|---|---|---|---|---|
| | | RO-16 | RB-5 | RO-16 | RB-5 | RO-16 | RB-5 |
| 50 | Free cells | 1.00 | 1.00 | 0.90 | 0.91 | 0.91 | 0.94 |
| | IMC | 1.00 | 1.00 | 0.91 | 0.91 | 0.91 | 0.89 |
| 100 | Free cells | 1.00 | 0.99 | 0.90 | 0.95 | 0.81 | 0.81 |
| | IMC | 0.99 | 0.99 | 0.93 | 0.97 | 0.86 | 0.97 |
| 150 | Free cells | 0.98 | 0.99 | 0.90 | 0.90 | 0.90 | 0.69 |
| | IMC | 0.98 | 0.99 | 0.90 | 0.98 | 0.80 | 0.97 |
| 200 | Free cells | 0.99 | 0.99 | 0.81 | 0.88 | 0.66 | 0.66 |
| | IMC | 0.97 | 0.99 | 0.88 | 0.98 | 0.76 | 0.69 |
| 250 | Free cells | 0.99 | 0.99 | 0.91 | 0.93 | 0.71 | 0.71 |
| | IMC | 0.99 | 0.99 | 0.91 | 0.8 | 0.72 | 0.61 |

found that the post-treatment (6 days) dye samples i.e., that retain the metabolites of the dyes, exhibit no toxic effects against all the three reference bacterial strains, while dyes at all tested concentrations (50 ppm, 150 ppm, 250 ppm) inhibited the growth of bacterial strains (Tables 4 and 5). The result of the present study clearly indicates that the *B. cereus* strain ROC can detoxify both dyes and the metabolites that are generated after treatment have no toxic effect on selected bacterial strains although the parent dyes significantly inhibited their growth. The antimicrobial activity with the untreated RO-16 and RB-5 at 250 ppm concentration sample could be due to the toxic nature of these dyes. A study carried out by Ajaz et al., (2019) reported that the *Alcaligenes aquatilis* 3c (bacterium) treated dye water was also found nontoxic for microbial growth [47]. Similarly, Phulpoto et al., (2018) reported the reduction in toxicity of post treated oil-based paint against bacterial and fungal strains [18].

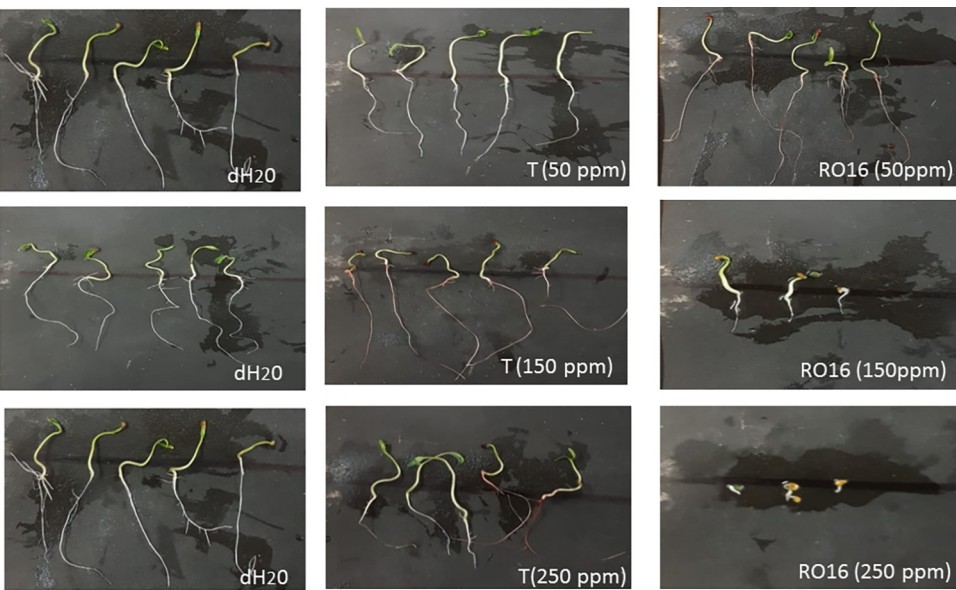

**Fig 6. Comparison of phytotoxicity on growth parameter of plant *Solanum lycopersicum* (*Riogrande*).** While treating it with various concentrations of reactive orange 16, treated reactive orange 16 with ROC (T) and control (dH₂O).

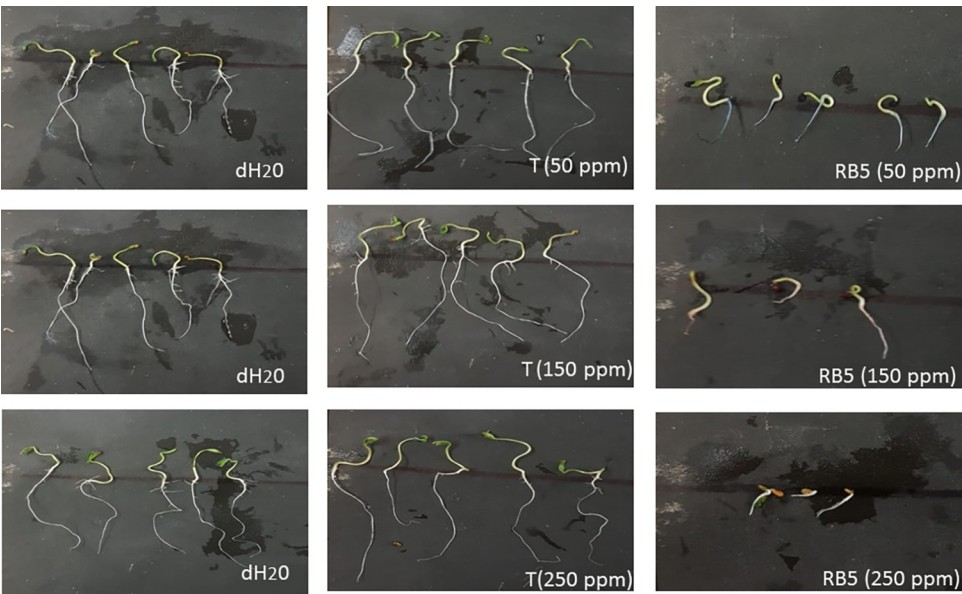

**Fig 7. Comparison of phytotoxicity on growth parameter of plant *Solanum lycopersicum (Riogrande*).** While treating it with various concentrations of reactive black 5, treated reactive orange 5 with ROC (T) and control (dH₂O).

## Conclusion

The present study was focused on bacterial degradation of the two most abundantly used azo dyes throughout the world. The bacterium *B. cereus* strain ROC is shown to be highly proficient in degrading both dyes i.e., RO-16 and RB-5. Growth independent kinetic models were applied which also revealed that the bacterium was able to degrade the azo dyes co-metabolically. Furthermore, immobilized cells of strain ROC degraded these dyes more efficiently as compared to their free cell counter parts. Thus, this study provides support for the practical application of *B. cereus* ROC in aerobic environments that are contaminated with azo dyes. Future work, will focus on the identification and characterization of the dye-degrading enzymes from the isolated bacterium.

**Table 2. Phytotoxicity comparison of Reactive orange 16 and its extracted metabolites.**

| Treatments | Shoot Length | Root length (cm) | Percentage Germination |
|---|---|---|---|
| Control (Water) | 0.9±0.01 a | 2.2±0.07 a | 92.6±0.7 a |
| **Pure dye RO-16** | | | |
| 50 ppm | 0.8±0.09 b | 1.7±0.02 c | 86.1±0.4 c |
| 150 ppm | 0.1±0.06 d | 0.1±0.01 e | 12.5±0.5 c |
| 250 ppm | 0.1±0.04 d | 0.1±0.02 e | 6.04±0.5 e |
| **Degraded dye** | | | |
| 50 ppm | 0.9±0.09 a | 1.8±0.02 b | 91.8±0.7 b |
| 150 ppm | 0.8±0.02 b | 1.7±0.06 c | 86.5±0.4 c |
| 250 ppm | 0.6±0.01 c | 1.0±0.01 d | 80.5±0.8 d |

Values are means of three replicates ± SD by two-way ANOVA $P < 0.005$ where Duncan test was applied within the groups.

**Table 3. Phytotoxicity comparison of Reactive black 5 and its extracted metabolites.**

| Treatments | Shoot length (cm) | Root length (cm) | Percentage germination |
|---|---|---|---|
| Control (Water) | 0.9±0.01 a | 2.2±0.07 a | 92.6±0.7 a |
| **Pure dye RB-5** | | | |
| 50 ppm | 0.8±0.02 b | 2.±0.01 b | 85.8±0.8 d |
| 150 ppm | 0.1±0.01 e | 0.4±0.5 d | 13±0.2 d |
| 250 ppm | 0.1±0.02 e | 0.1±0.01 e | 6.1±0.4 e |
| **Degraded dye** | | | |
| 50 ppm | 0.8±0.01 b | 2.1±0.09 a | 91.8±0.7 b |
| 150 ppm | 0.7±0.01 c | 2.±0.12 b | 86.5±0.4 c |
| 250 ppm | 0.6±0.02 d | 1.1±0.06 c | 85.8±0.8 d |

Values are means of three replicates ± SD by two-way ANOVA $P < 0.005$ where Duncan test was applied within the groups.

**Table 4. Antimicrobial activity of treated and untreated Reactive black 5 samples against bacteria.**

| Tested Bacterial strains | Reactive Black 5 | | | | | |
|---|---|---|---|---|---|---|
| | Antimicrobial effect of treated vs. untreated dye samples (zone of inhibition in mm±Std. Dev.) | | | | | |
| | 0 day (untreated samples) | | | 6 days (treated samples) | | |
| | 50 ppm | 150 ppm | 250 ppm | 50 ppm | 150ppm | 250ppm |
| *Pseudomonas aurignosa* | 8.0±0.5 h | 11.0±0.2 e | 13.0±0.5 c | 0.0±0.0 i | 0.0±0.0 i | 0.0±0.0 i |
| *E.coli* | 9.0±0.3 g | 11.0±0.2 e | 14.0±0.2 b | 0.0±0.0 i | 0.0±0.0 i | 0.0±0.0 i |
| *Staphyloccous aureus* | 8.0±0.2 h | 12.0±0.3 d | 13.0±0.3 c | 0.0±0.0 i | 0.0±0.0 i | 0.0±0.0 i |
| **Positive Control** | 10.0±0.5 f | 15.0±0.5 a | 15.0±0.3 a | 0.0±0.0 i | 0.0±0.0 i | 0.0±0.0 i |
| **Negative Control** | 0.0±0.0 i | 0.0±0.0 i | 0.0±0.0 i | 0.0±0.0 i | 0.0±0.0 i | 0.0±0.0 i |

Positive control used is antibiotic (ampicillin) and negative control used is water. Different alphabets show the significant differences ($p < 0.05$) between the treatments.

**Table 5. Antimicrobial activity of treated and untreated Reactive orange 16 samples against bacteria.**

| Tested Bacterial strains | Reactive orange 16 | | | | | |
|---|---|---|---|---|---|---|
| | Antimicrobial effect of treated vs. untreated dye samples (zone of inhibition in mm±Std. Dev.) | | | | | |
| | 0 day (untreated samples) | | | 6 days (treated samples) | | |
| | 50 ppm | 150 ppm | 250 ppm | 50 ppm | 150 ppm | 250 ppm |
| *Pseudomonas aurignosa* | 9.0±0.5 g | 11.0±0.2 f | 13.0±0.5 d | 0.0±0.0 i | 0.0±0.0 i | 0.0±0.0 i |
| *E.coli* | 9.5±0.3 g | 11.3±0.2 f | 14.0±0.2 c | 0.0±0.0 i | 0.0±0.0 i | 0.0±0.0 i |
| *Staphyloccous aureus* | 8.0±0.2 h | 12.0±0.3 e | 13.0±0.3 d | 0.0±0.0 i | 0.0±0.0 i | 0.0±0.0 i |
| **Positive Control** | 10.0±0.5 g | 16.0±0.5 a | 15.0±0.3 b | 0.0±0.0 i | 0.0±0.0 i | 0.0±0.0 i |
| **Negative Control** | 0.0±0.0 i | 0.0±0.0 i | 0.0±0.0 i | 0.0±0.0 i | 0.0±0.0 i | 0.0±0.0 i |

Positive control used is antibiotic (ampicillin) and negative control used is water. Different alphabets show the significant differences ($p < 0.05$) between the treatments.

## Supporting information

**S1 Fig.** Decolourization of Reactive orange 16 by free and immobilized cells of bacterial strain (A) by free cells at 30 ˚C (B) by immobilized cells at 30 ˚C.
(TIF)

**S2 Fig.** Decolorization of Reactive black 5 by free and immobilized cells of bacterial strain (A) by free cells at 30 ˚C (B) by immobilized cells at 30 ˚C.
(TIF)

**S3 Fig.** Concentrations of Reactive orange 16 degraded after treatment with free and immobilized cells of the bacterial strain (A) Degraded by free cells at 30 ˚C (B) Degraded by immobilized cells at 30 ˚C.
(TIF)

**S4 Fig.** Concentrations of Reactive black 5 degraded after treatment with free and immobilized cells of the bacterial strain (A) Degraded by free cells at 30 ˚C (B) Degraded by immobilized cells at 30 ˚C.
(TIF)

## Author Contributions

**Conceptualization:** Tatheer Alam Naqvi.

**Data curation:** Anum Fareed, Habiba Zaffar, Tatheer Alam Naqvi.

**Formal analysis:** Anum Fareed, Habiba Zaffar, Muhammad Bilal, Jamshaid Hussain, Tatheer Alam Naqvi.

**Funding acquisition:** Colin Jackson.

**Investigation:** Anum Fareed, Tatheer Alam Naqvi.

**Methodology:** Habiba Zaffar, Jamshaid Hussain, Tatheer Alam Naqvi.

**Project administration:** Tatheer Alam Naqvi.

**Resources:** Muhammad Bilal, Tatheer Alam Naqvi.

**Software:** Habiba Zaffar, Muhammad Bilal, Jamshaid Hussain.

**Supervision:** Colin Jackson, Tatheer Alam Naqvi.

**Validation:** Colin Jackson, Tatheer Alam Naqvi.

**Visualization:** Jamshaid Hussain, Colin Jackson, Tatheer Alam Naqvi.

**Writing – original draft:** Anum Fareed, Tatheer Alam Naqvi.

**Writing – review & editing:** Colin Jackson, Tatheer Alam Naqvi.

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
