## [Decision Letter · Decision Letter 0]

17 Mar 2022

PONE-D-22-01377Decolorization of Azo dyes by a novel aerobic bacterialPLOS ONE

Dear Dr. Naqvi,

Thank you for submitting your manuscript to PLOS ONE. After careful consideration, we feel that it has merit but does not fully meet PLOS ONE’s publication criteria as it currently stands. Therefore, we invite you to submit a revised version of the manuscript that addresses the points raised during the review process.

We look forward to receiving your revised manuscript.

Kind regards,

Faisal Mamhood

Academic Editor

PLOS ONE

Journal Requirements:

3. In your Methods section, please provide additional location information about your sampling sites, including geographic coordinates for the data set if available.

4. In your Methods section, please provide additional information regarding the permits you obtained for the work. Please ensure you have included the full name of the authority that approved the field site access and, if no permits were required, a brief statement explaining why.

Additional Editor Comments:

Please find below the reviewers comments 

Reviewers' comments:

Reviewer's Responses to Questions

**Comments to the Author**

1. Is the manuscript technically sound, and do the data support the conclusions?

Reviewer #1: Yes

Reviewer #2: Yes

2. Has the statistical analysis been performed appropriately and rigorously? 

Reviewer #1: No

Reviewer #2: Yes

3. Have the authors made all data underlying the findings in their manuscript fully available?

Reviewer #1: Yes

Reviewer #2: Yes

4. Is the manuscript presented in an intelligible fashion and written in standard English?

Reviewer #1: No

Reviewer #2: Yes

5. Review Comments to the Author

Reviewer #1: The manuscript titled "Decolorization of azo dyes by a novel bacterial' has been reviewed and I found it an interesting piece of research; however, a lot of flaws have been indicated and are necessary to be taken care of before the further consideration.

Comments:

The title of the paper is not complete ‘bacterial isolate………..name…..?? What is the novel in the bacterial strain, a huge number of bacillus species have been reported to decolorize the dyes. The word Novel is confusing here.

Abstract………azo dyes…not azo- dyes

Introduction contain a lot of grammar and structural mistakes. Please take care of…

Methodology

All chemicals and reagents of analytical grade…..this is not a complete sentence

What is the rational to select 37 C temperature and why the aerobic conditions were provided especially when it is established that the azo dye decolorization is an anaerobic process?

16S rRNA gene….should be written correctly….Which primers were used?.....the mention the protocol description

Phylogenetic tree should be correctly drawn. All the bacterial strain should be italicized. The isolated bacterial strain should be indicated as bold or red. A control group should be added in the tree

In decolorization graphs, if decolorization was not measured at 0, then 0 time should not be added in the graphs. There is no need to add 0

There should be an accession number of the identified strain

The discussion is all over the place. The mere repetition of results should not be the part. Discussion should logically narrate the results by comparting with relevant literature.

The manuscript should be checked with the native English speaker to improve the grammar, syntax, sentence structure etc.

Reviewer #2: This is a well designed and clearly written study. The authors have performed that experiments in a logical manner and reported the results with comparison to the available studies in the literature. The quality of English language is also adequate. Therefore, this study has my recommendation for publication as such.

I have a minor question though. Did the authors test the growth potential of the isolated strain against the azo dyes at concentrations levels higher than the 250 mg/l? If not, why?

6. PLOS authors have the option to publish the peer review history of their article (what does this mean?). If published, this will include your full peer review and any attached files.

Reviewer #1: No

Reviewer #2: No

---

## [Author Response · Author response to Decision Letter 0]

22 Apr 2022

Respond to Reviewers

Manuscript was checked thoroughly and follow the instructions provided by the journal site. 

2. We suggest you thoroughly copyedit your manuscript for language usage, spelling, and grammar. If you do not know anyone who can help you do this, you may wish to consider employing a professional scientific editing service

Professor Colin Jackson who is a coauthor in this research article, has reviewed this manuscript for scientific as well as for English errors, whose native language is English.

3. In your Methods section, please provide additional location information about your sampling sites, including geographic coordinates for the data set if available.

The required information is given on Page #7, line # 145-146.

4. In your Methods section, please provide additional information regarding the permits you obtained for the work. Please ensure you have included the full name of the authority that approved the field site access and, if no permits were required, a brief statement explaining why.

The information is given on page #7, line # 146-148.

Reviewer #1: The manuscript titled "Decolorization of azo dyes by a novel bacterial' has been reviewed and I found it an interesting piece of research; however, a lot of flaws have been indicated and are necessary to be taken care of before the further consideration.

Comments:

1. The title of the paper is not complete ‘bacterial isolate………..name…..?? What is the novel in the bacterial strain, a huge number of bacillus species have been reported to decolorize the dyes. The word Novel is confusing here.

Answer: The name of the bacterial isolate is added in the title Page # 1, 2, line # 2, 23. The word novel is used for this strain as it decolorize the dyes aerobically.

2. Abstract………azo dyes…not azo- dyes

Answer: It has been corrected in the abstract. Page # 2, line # 29

3. Introduction contain a lot of grammar and structural mistakes. Please take care of…

Answer: 

Methodology

4. All chemicals and reagents of analytical grade…..this is not a complete sentence 

Answer: The sentence has been revised. Page # 7 , line #119

5. What is the rational to select 37 C temperature and why the aerobic conditions were provided especially when it is established that the azo dye decolorization is an anaerobic process?

Answer: Optimization of azo dyes degradation was done at various temperature i.e., 25,32,37ºC and 45ºC. but maximum decolorization with short period of time was observed at 37ºC. 

Although the azo dyes are extremely stable under aerobic conditions and the spontaneous reduction of the azo bond under anaerobic conditions leads to highly carcinogenic and mutagenic colorless aromatic amines so the present study focused on the degradation of azodyes under aerobic conditions. 

6. 16S rRNA gene….should be written correctly….Which primers were used?.....the mention the protocol description

Answer: The protocol has been described. Page # 8, Line #143, 144, 145

7. Phylogenetic tree should be correctly drawn. All the bacterial strain should be italicized. The isolated bacterial strain should be indicated as bold or red. A control group should be added in the tree

Answer:

8. In decolorization graphs, if decolorization was not measured at 0, then 0 time should not be added in the graphs. There is no need to add 0

Answer: The decolorization was also measured at 0, that’s why the 0 time is added in the graphs.

9.There should be an accession number of the identified strain

Answer: The accession number of the strain has been added. Page # 12, Line # 232. 

10. The discussion is all over the place. The mere repetition of results should not be the part. Discussion should logically narrate the results by comparting with relevant literature.

Answer:

11. The manuscript should be checked with the native English speaker to improve the grammar, syntax, sentence structure etc.

Answer:

Reviewer #2: This is a well designed and clearly written study. The authors have performed that experiments in a logical manner and reported the results with comparison to the available studies in the literature. The quality of English language is also adequate. Therefore, this study has my recommendation for publication as such.

1. I have a minor question though. Did the authors test the growth potential of the isolated strain against the azo dyes at concentrations levels higher than the 250 mg/l? If not, why?

Answer: The growth potential of the isolated strain against the azo dyes at concentrations levels higher than 250 mg/l was also tested but no degradation was observed it is not mentioned in the article.

---

## [Editor Report · Decision Letter 1]

17 May 2022

PONE-D-22-01377R1Decolorization of Azo dyes by a novel aerobic bacterial strain Bacillus cereus strain ROCPLOS ONE

Dear author

Thank you for submitting your manuscript to PLOS ONE. After careful consideration, we feel that it has merit but does not fully meet PLOS ONE’s publication criteria as it currently stands. Therefore, we invite you to submit a revised version of the manuscript that addresses the points raised during the review process.

Please include the following items when submitting your revised manuscript:A rebuttal letter that responds to each point raised by the academic editor and reviewer(s). You should upload this letter as a separate file labeled 'Response to Reviewers'.A marked-up copy of your manuscript that highlights changes made to the original version. You should upload this as a separate file labeled 'Revised Manuscript with Track Changes'.An unmarked version of your revised paper without tracked changes. You should upload this as a separate file labeled 'Manuscript'.If applicable, we recommend that you deposit your laboratory protocols in protocols.io to enhance the reproducibility of your results. Protocols.io assigns your protocol its own identifier (DOI) so that it can be cited independently in the future. For instructions see: https://journals.plos.org/plosone/s/submission-guidelines#loc-laboratory-protocols. Additionally, PLOS ONE offers an option for publishing peer-reviewed Lab Protocol articles, which describe protocols hosted on protocols.io. Read more information on sharing protocols at https://plos.org/protocols?utm_medium=editorial-email&utm_source=authorletters&utm_campaign=protocols.

We look forward to receiving your revised manuscript.

Kind regards,

Faisal Mamhood

Academic Editor

PLOS ONE

Journal Requirements:

Additional Editor Comments (if provided):

Some answers are not given in the revised version, e.g, question raised by reviewer 1 as mentioned by the author as question number 10 and 11 in file response to reviewers comments. Give the answers of these questions.

Moreover make a separate file regarding response to reviewers comments for both reviewers, in which question raised by reviewers and answer of that question should be clearly readable

In the manuscript also highlight the text which is added regarding the answers of the reviewers comments throughout the manuscript
---

## [Author Response · Author response to Decision Letter 1]

18 May 2022

Respond to Reviewers

Manuscript was checked thoroughly and follow the instructions provided by the journal site. 

2. We suggest you thoroughly copyedit your manuscript for language usage, spelling, and grammar. If you do not know anyone who can help you do this, you may wish to consider employing a professional scientific editing service

Professor Colin Jackson who is a coauthor in this research article, has reviewed this manuscript for scientific as well as for English errors, whose native language is English.

3. In your Methods section, please provide additional location information about your sampling sites, including geographic coordinates for the data set if available.

The required information is given on Page #7, line # 145-146.

4. In your Methods section, please provide additional information regarding the permits you obtained for the work. Please ensure you have included the full name of the authority that approved the field site access and, if no permits were required, a brief statement explaining why.

The information is given on page #7, line # 146-148.

Reviewer #1: The manuscript titled "Decolorization of azo dyes by a novel bacterial' has been reviewed and I found it an interesting piece of research; however, a lot of flaws have been indicated and are necessary to be taken care of before the further consideration.

Comments:

1. The title of the paper is not complete ‘bacterial isolate………..name…..?? What is the novel in the bacterial strain, a huge number of bacillus species have been reported to decolorize the dyes. The word Novel is confusing here.

Answer: The name of the bacterial isolate is added in the title Page # 1, 2, line # 2, 23. The word novel is used for this strain as it decolorize the dyes aerobically.

2. Abstract………azo dyes…not azo- dyes

Answer: It has been corrected in the abstract. Page # 2, line # 29

3. Introduction contain a lot of grammar and structural mistakes. Please take care of…

Answer: 

Methodology

4. All chemicals and reagents of analytical grade…..this is not a complete sentence 

Answer: The sentence has been revised. Page # 7 , line #119

5. What is the rational to select 37 C temperature and why the aerobic conditions were provided especially when it is established that the azo dye decolorization is an anaerobic process?

Answer: Optimization of azo dyes degradation was done at various temperature i.e., 25,32,37ºC and 45ºC. but maximum decolorization with short period of time was observed at 37ºC. 

Although the azo dyes are extremely stable under aerobic conditions and the spontaneous reduction of the azo bond under anaerobic conditions leads to highly carcinogenic and mutagenic colorless aromatic amines so the present study focused on the degradation of azodyes under aerobic conditions. 

6. 16S rRNA gene…should be written correctly….Which primers were used?.....the mention the protocol description

Answer: The protocol has been described. Page # 8, Line #143, 144, 145

7. Phylogenetic tree should be correctly drawn. All the bacterial strain should be italicized. The isolated bacterial strain should be indicated as bold or red. A control group should be added in the tree

Answer:

8. In decolorization graphs, if decolorization was not measured at 0, then 0 time should not be added in the graphs. There is no need to add 0

Answer: The decolorization was also measured at 0, that’s why the 0 time is added in the graphs.

9.There should be an accession number of the identified strain

Answer: The accession number of the strain has been added. Page # 12, Line # 232. 

10. The discussion is all over the place. The mere repetition of results should not be the part. Discussion should logically narrate the results by comparting with relevant literature.

Answer: Discussion has been improved as according to the suggestions made by the reviewer (Section: Results and Discussion). But as the section is Results along with discussion so we make try our best to not repeat results during discussion and solely discuss results after showing the results of the experiments.

11. The manuscript should be checked with the native English speaker to improve the grammar, syntax, sentence structure etc.

Answer: Professor Dr. Colin Jackson who belongs to Australia (Native English speaker) has checked the whole manuscript and improve the language as well as technical errors if there is any. He has read the manuscript thoroughly and incorporate the changes as shown in Revised Manuscript with Track Changes

Reviewer #2: This is a well-designed and clearly written study. The authors have performed that experiments in a logical manner and reported the results with comparison to the available studies in the literature. The quality of English language is also adequate. Therefore, this study has my recommendation for publication as such.

1. I have a minor question though. Did the authors test the growth potential of the isolated strain against the azo dyes at concentrations levels higher than the 250 mg/l? If not, why?

Answer: The growth potential of the isolated strain against the azo dyes at concentrations levels higher than 250 mg/l was also tested but no degradation was observed it is not mentioned in the article.

---

## [Editor Report · Decision Letter 2]

24 May 2022

Decolorization of Azo dyes by a novel aerobic bacterial strain Bacillus cereus strain ROC

PONE-D-22-01377R2

Dear authors

We’re pleased to inform you that your manuscript has been judged scientifically suitable for publication and will be formally accepted for publication once it meets all outstanding technical requirements.

Kind regards,

Faisal Mamhood

Academic Editor

PLOS ONE
---

## [Editor Report · Acceptance letter]

30 May 2022

PONE-D-22-01377R2 

Decolorization of Azo dyes by a novel aerobic bacterial strain *Bacillus cereus* strain ROC 

Dear Dr. Naqvi:

I'm pleased to inform you that your manuscript has been deemed suitable for publication in PLOS ONE. Congratulations! Your manuscript is now with our production department. 

Kind regards, 

on behalf of

Dr. Faisal Mamhood 

Academic Editor

PLOS ONE